# Interpretable Cross-Subject EEG-Based Emotion Recognition Using Channel-Wise Features^†^

**DOI:** 10.3390/s20236719

**Published:** 2020-11-24

**Authors:** Longbin Jin, Eun Yi Kim

**Affiliations:** Computer Science and Engineering, Konkuk University, Seoul 05029, Korea; jinlongbin@konkuk.ac.kr

**Keywords:** EEG, cross-subject, emotion recognition, user independent model, channel-wise feature

## Abstract

Electroencephalogram (EEG)-based emotion recognition is receiving significant attention in research on brain-computer interfaces (BCI) and health care. To recognize cross-subject emotion from EEG data accurately, a technique capable of finding an effective representation robust to the subject-specific variability associated with EEG data collection processes is necessary. In this paper, a new method to predict cross-subject emotion using time-series analysis and spatial correlation is proposed. To represent the spatial connectivity between brain regions, a channel-wise feature is proposed, which can effectively handle the correlation between all channels. The channel-wise feature is defined by a symmetric matrix, the elements of which are calculated by the Pearson correlation coefficient between two-pair channels capable of complementarily handling subject-specific variability. The channel-wise features are then fed to two-layer stacked long short-term memory (LSTM), which can extract temporal features and learn an emotional model. Extensive experiments on two publicly available datasets, the Dataset for Emotion Analysis using Physiological Signals (DEAP) and the SJTU (Shanghai Jiao Tong University) Emotion EEG Dataset (SEED), demonstrate the effectiveness of the combined use of channel-wise features and LSTM. Experimental results achieve state-of-the-art classification rates of 98.93% and 99.10% during the two-class classification of valence and arousal in DEAP, respectively, with an accuracy of 99.63% during three-class classification in SEED.

## 1. Introduction

Emotions are fundamental in the daily lives of humans, and they play an essential role in decision-making, human interactions, and even mental health [1]. For instance, in the medical field of psychiatry, detected emotional states of patients help identify those at a high risk of emotional disorders and depression [2]. Thus, there has been much research on emotion recognition using facial expressions [3], thermography [4], motion capture system [5], text [6], and speech [7]. However, these modes are difficult for representing people’s true feelings because they are sensitive to subject-specific variability. Moreover, people can express false emotions.

To solve this problem, electroencephalogram (EEG) has been considered as an alternative for detecting emotions produced unintentionally by the human brain. As a typical central nervous signal, the EEG signal directly reflects the strength and position of brain activity at a high temporal resolution [8]. Therefore, EEG signals are more stable for extracting the actual emotional states of humans. Benefiting from many non-invasive and easy-to-wear EEG measuring devices, it is easy to monitor electrical brain activity with EEG. Due to these advantages, EEG-based research has been relatively active.

EEG-based emotion recognition model design can follow a user-dependent or a user-independent approach. In the case of a user-dependent model, training and testing data are chosen from the same subject. Therefore, an emotion recognition model typically shows high accuracy on a user-dependent model. However, such a user-dependent model lacks generalization and a tuning process is necessary for each new subject, which requires training data from each subject. Thus, it is desirable to develop a user-independent model. In this scenario, a recognition system is trained using data from some subjects while applied to new subjects in testing. In contrast, a user-independent model is more applicable to new users because there is no need to create a new model [9].

The main issue about recognizing cross-subject emotion from EEG signals is to find effective representations that are robust to subject-specific variability and noise associated with the EEG data collection process. EEG signals have a low signal-to-noise ratio (SNR) and are affected by common noise pattern of sensor systems, as well as unintentional physical activities such as eye blinks and muscle movement, which make it difficult to recognize the emotion states from raw EEG signals. Moreover, due to the subject-specific variability, it is difficult to find invariant emotion-related features from different subjects. To handle these problems, various emotion-related feature extraction methods have been developed. These methods can be sorted into two categories: human crafted feature-based approaches and deep feature-based ones.

The most common methods to recognize human emotion from EEG signals have been relying on some hand-crafted features. Some methods extracted delta, theta, alpha, beta, and gamma waves using a bandpass filter [10], and other methods implemented the wavelet transform (WT) to extract emotion-related features [11]. In addition, researchers focused on investigating critical emotion-related frequency bands and channels. Zheng et al. found that different emotions have different emotion-related bands and channels [12]. Although these signal processing methods can explicitly suppress noise and artifacts, they did not consider the subject-specific variability of EEG.

Recently, deep learning techniques are applied to automatically model brain activity. Moreover, with the discovery of the spatial connectivity of EEG, many studies have begun to combine the spatial connectivity of different brain regions with the temporal change of EEG signals, for a more accurate emotion recognition model. Yang et al. transformed the data into topology-preserving two-dimensional (2D) EEG frames based on the International 10–20 system [13]. Then, the 2D matrices were input to a parallel convolutional recurrent neural network to learn the spatial and temporal representation separately. Wang et al. reshaped raw EEG data into three-dimensional (3D) tensors (2D electrode topological structure × time samples) and used a 3D CNN architecture, named EmotioNet, to extract the spatial and temporal features simultaneously [14].

The hand-crafted feature-based approaches can explicitly reduce the noise and find emotion-related features. However, they rarely consider the subject-specific variability and usually required specified domain knowledge to extract hand-crafted features. By deep learning methods, the relevant features of the emotions are automatically extracted from the raw EEG signals. Generally, most of the works only report the results achieved by deep learning, without detailed explanations or insights about the results. Besides presenting the classification performance, it is also important to interpret the cause of such classification success.

To overcome the mentioned limitations of feature representation methods, we present an interpretable cross-subject EEG-based emotion recognition model using the combination of hand-crafted features and a deep learning approach. More specifically, we extract channel-wise features that integrate spatial connectivity of whole brain regions and use LSTM to learn temporal information. The channel-wise feature is defined by a symmetric matrix and considers the linear combination of every two-pair channels. In this way, the channel-wise feature is enabled to encode the features of individuals, and capable of complementarily handling subject-specific variability. We include visualizations of channel-wise features and show that the channel-wise features are robust to subject-specific variability. Then, to reduce the bias of subject-specific variability, a sequence of channel-wise features is fed to two-layer stacked LSTM layers. We allow the LSTM layer to automatically learn emotional features for discriminating between emotion type.

The effectiveness of our model was examined on two publicly accessible datasets, specifically the Dataset for Emotion Analysis using Physiological Signals (DEAP) [15] and the SJTU (Shanghai Jiao Tong University) Emotion EEG Dataset (SEED) [16]. For the DEAP, our model achieves state-of-the-art accuracy of 98.93% and 99.10% on two-class (high, low) valence and arousal classification tasks, respectively, and achieves 98.32% on four-class (high valence high arousal, high valence low arousal, low valence high arousal, low valence low arousal) classification in one model. For the SEED, emotion classification with three classes (positive, neutral, negative) achieves an accuracy of 99.63%.

Our contributions are summarized as follows:We propose a cross-subject EEG-based emotion recognition model using a combination of channel-wise features and LSTM. The channel-wise features consider the spatial connectivity of whole brain regions, which have robust subject-specific variability, and LSTM can learn the temporal information and extract the emotion-related feature.We implement extensive experiments in both the DEAP and SEED and carry out a systematic comparison with different studies. Experimental results outperform the state of the art by a large margin and demonstrate the effectiveness of the proposed model.We investigate the properties of channel-wise features and experimentally demonstrate that the presented channel-wise features can reduce negative effects due to subject-specific variability.

The rest of this paper is organized as follows: Section 2 begins by introducing the previous research of EEG-based emotion recognition and provides an understanding of some basic emotional feature extraction concepts such as hand-crafted features and deep features. Section 3 summarizes the entire process of our model. A detailed description of the proposed method and LSTM structure is presented in Section 4. Section 5 contains the detailed information of the DEAP and SEED, experimental setting and results to demonstrate the effectiveness of our model. Finally, the main conclusion of our research is presented in Section 6.

## 2. Related Works

Until now, many studies have focused on finding emotion-related features from EEG raw signals and developing classification methods using the extracted features. Some methods have tried to extract the crucial features on a single EEG channel by frequency decomposition, and others have tried to combine the spatial information and temporal information from raw signals using deep learning techniques directly. Thus, the EEG-based emotion representation methods can mainly be divided into two approaches: EEG representation on the time-frequency domain and spatiotemporal domain.

### 2.1. EEG Representation on Time-Frequency Domain

Traditionally, the most challenging task of extract EEG representation on the time-frequency domain is to remove noise from raw EEG signals and to find the emotion-related features without damaging the signals related to emotion. The EEG signals are divided into five bands: delta (0~4 Hz), theta (4~7.5 Hz), alpha (7.5~12.5 Hz), beta (12.5~30 Hz), and gamma (30~40 Hz) [10,17,18]. Among them, beta and gamma are more associated with brain activity (relevant to the active state of the mind). The most used method to compute power spectral features are Fourier transform [10] and wavelet transform [11,17]. Koelstra et al. developed the DEAP and extracted power spectral features from all 32 electrodes and achieve the emotion classification accuracy of 62.0% for two-class arousal classification and 57.6% for two-class valence classification using a user-independent model [15]. Li et al. explored the robustness of a wider range of EEG features, including nine kinds of time-frequency domain features and nine kinds of dynamical system features from EEG measurements [19]. These features were fed to the support vector machine (SVM) and achieve an accuracy of 59.06% on the DEAP and of 83.33% on the SEED using a user-independent model. Candra et al. extracted wavelet entropy features from the alpha, beta, gamma band and classified the emotion using SVM [20]. They also refined the result by using 10 different window sizes of EEG segments. The result showed that 3–12 s window is most suitable for emotion recognition. Based on this study, Tripathi et al. segmented EEG data into 6 s range and extracted the mean, median, maximum, minimum, standard deviation, variance, range, skewness, and kurtosis values from each segment [21]. These features were then input into DNN for emotion classification and achieve the accuracy of 75.78% and 73.125% on two-class valence and arousal classification using user-independent model, respectively.

There is growing interest in finding the emotion-related EEG channels and bands. Zheng et al. developed SEED and use deep belief networks (DBNs) to classify three emotions [16]. By training the DBNs, they found the four different profiles of 4, 6, 9, and 12 channels as critical channels, beta, and gamma bands of EEG data are more related to emotion recognition. Gupta et al. investigated the channel-specific nature of EEG signals based on a flexible analytic wavelet transform (FAWT) for recognizing cross-subject emotions [22]. They also used 12 channels in SEED and 6 channels in DEAP, which are more suitable for emotion recognition as suggested in [16]. Zheng et al. also found that stable neural patterns of positive, neutral, and negative emotions [12]. The positive emotions have higher beta and gamma responses at the lateral temporal areas, parietal and occipital sites are more active for neutral emotions in the alpha band, and the negative emotion patterns have higher delta responses at parietal and occipital sites and higher gamma response at prefrontal sites.

The abovementioned methods consist of two main modules: extracting hand-crafted features and classifying emotions using machine learning techniques such as SVM, DBNs, and so on. Those methods can effectively reduce the artifacts included in raw EEG signals and they used lightweight learning models. Additionally, they investigated some features strongly related to some emotions and showed the effectiveness of such emotion-specific features in the stage of classification. However, they show a limitation in improving overall performance for all emotional categories. It remains a challenging issue to identify features that are strongly related to all of the emotions. 

### 2.2. EEG Representation on Spatiotemporal Domain

Tripathi et al. used a convolutional neural network (CNN) for the first time to classify the classes of valence and arousal in the DEAP [21]. A CNN can capture the spatial dependencies of EEG features through the utilization of relevant filters. Wang et al. reshaped raw EEG data into 3D tensors and used a 3D CNN architecture to extract the spatial and temporal features simultaneously [14]. They then implemented a dense prediction approach to provide the network with general emotional information more efficiently. Kim et al. applied convolutional LSTM to capture local dependencies in the spatiotemporal domain [23]. Although this method can extract spatiotemporal features automatically, the accuracy was lower than 80% in the user-independent model because it did not consider subject-specific variability. Yang et al. found BaseMean, which is used to represent subjects’ basic emotional states, subtracting the BaseMean outcome from raw EEG data [13]. The preprocessed data are then converted to 2D EEG frames based on the International 10–20 system, and the 2D matrices are input into a parallel convolutional recurrent neural network to learn the spatial and temporal representation separately. The accuracy of valence and arousal on the DEAP exceeded 90% in a user-dependent model.

The graph theory approaches using Graph Neural Networks (GNN) to characterize the brain connectivity are getting more attention. Song et al. presented a dynamical graph convolutional neural networks (DGCNN) to model the multichannel EEG features and perform EEG emotion classification [24]. This method can dynamically learn the spatial connectivity between different brain regions, represented by an adjacency matrix by training the networks, and the user-independent emotion recognition accuracy was 79.95% for the three-class valence on the SEED. Zhong et al. reported a regularized graph neural network (RGNN), which captures both local and global inter-channel relations [25]. The inter-channel relations are shown by the adjacency matrix, where the connection and sparseness are supported by the neuroscience theories of the human brain structure. Although these methods can extract the connectivity between brain regions automatically, they did not consider the subject-specific variability since the relevant channels vary from person to person.

As can be seen from the abovementioned methods, user-dependent models provide accurate recognition performance, while most user-independent models show low recognition performance. However, the user-dependent models lack generalization, so they should generate an individual recognition model for each new user. Therefore, the user-independent model is more applicable to new users because there is no need to create a new model [9]. However, the performance of the user-independent model is not high enough to be used in a variety of practical applications. For example, to the best of our knowledge, the recognition rate of the user-independent model with the highest performance in the DEAP is less than 80%.

Until now, CNN based methods have achieved success in generating features important to emotion classification from raw EEG data. They can effectively extract not only single channel information but also the spatial correlation between adjacent channels through the convolution mechanism. However, this interaction is only investigated between physically adjacent channels.

Because the relevant channels vary from person to person, it is important to consider the connectivity between whole brain regions so as to reduce subject-specific variability.

## 3. Overview of the Proposed Method

We propose a novel EEG-based emotion recognition model that considers subject-specific variability in predictions of the emotions of a user omitted from the training set. As we have argued above, the main issue centers on how to identify the features that are strongly related to human emotions. We believe that the spatial connectivity between whole brain regions is an important clue in finding the emotion-specific features as well as subject-specific features. From this assumption, we first transform raw EEG signals to channel-wise features that can effectively represent distinctive connectivity patterns. It is assumed that a channel-wise feature can be separated into subject-specific patterns and emotion-specific patterns. Therefore, by filtering the subject-specific patterns from channel-wise features, only emotion-specific patterns can remain, which are used for emotion classification. In this work, LSTM is employed to extract an emotion-specific pattern by modeling the temporal dynamic behavior of channel-wise features.

A flowchart of our model is shown in Figure 1. First, we extracted single-channel features from raw EEG data to reduce the data size. Second, we extracted channel-wise features to model the spatial structure in neural correlations. Moreover, channel-wise features can explicitly model interdependencies between all channels, and this method can determine the unique pattern of each user’s EEG signal, allowing subject-specific variability to be considered. By successively extracting channel-wise features from several time steps, which are flattened and input into long short-term memory (LSTM), we can predict the emotions of users effectively. More formally, our model takes a sequence of raw EEG data E∈ℝC×L×K×N (Figure 1a) as input, where C is the number of EEG channels, L is the number of EEG data samples in each segment for each channel, K is the number of segments needed for considering the correlation of all EEG channels and extracting the channel-wise feature, and N is the number of time steps in the LSTM to extract the temporal emotional feature. Firstly, our model calculated single-channel features S∈ℝC×K×N (Figure 1b) by the dimension reduction from each L EEG data. Secondly, by considering the spatial connectivity between pairwise EEG channels from K single-channel feature values per channel, our model generates N channel-wise features F=F1,F2,…,FN (Figure 1c), where Fi∈ℝC×C. Then, we flatten the upper triangle of the channel-wise features (Figure 1d) and input the data into LSTM (Figure 1e). By training the parameters of LSTM, we can predict the emotional state accurately (Figure 1f).

## 4. Proposed Method

### 4.1. Preprocessing

Let T be the optimal times to obtain the EEG data of an individual and let R represent the sampling rate of the EEG signal. In the case of the DEAP, 32 EEG channels (C=32) were recorded at 128 Hz (R=128). Therefore, as many studies used one second EEG signal to recognize emotion [13,22], a one-second (T = 1) EEG signal of an individual has a data size of 32 × 128 × 1. If the raw data are input into the model, it will incur a large computational cost.

In order to reduce the data size of the EEG signals, we calculate the mean value of a segmented window from each EEG channel, which we term single-channel features. The input of our model is a sequence of EEG data E∈ℝC×L×K×N that contains C channels and L×K×N data per channel. All C channels are initially cut into K×N segments of the same length L. Hence, we obtain C×K×N segments, after which we calculate the mean value of the data within each segment. The value of the single-channel feature can be formulated as:(1)Sk,nc=meanEk,nc
where c∈1,C, k∈1,K, n∈1,N, and Ek,nc∈ℝL denote the input EEG data from the cth channel and the k, nth segment. We then obtain the S∈ℝC×K×N matrix, termed the single-channel feature.

The top of Figure 1 shows the input EEG signal and extracted single-channel features from the first subject in the DEAP. Each polyline represents a change in the trend of the single-channel features of one channel. All channel positions correspond to the 10–20 system. As can be observed in Figure 1, some channels have similar data movement trends over time. For example, Fp1, AF3, and F3 change with similar patterns, while P8, PO4, and O2 have similar patterns. On the other hand, certain channels move conversely. For example, F7 and P8 and AF3 and P4 move precisely opposite to each other. This indicates that some of the channels have high correlations with others, some of the channels have low interdependency rates, and other channels have no such relationships.

### 4.2. Channel-Wise Features

To observe such spatial connectivity between brain regions, we undertake a preliminary study using the DEAP. Among 32×32 pair-wise relations, three examples extracted from different subjects are shown in Figure 2. The data in Figure 2 are from Subject 1, 2, 3, and 4 in the DEAP when they feel high valence high arousal emotion for 60 s. The number below each scatter plot is the correlation coefficient; it is observed that a correlation does exist between the pair of channels and that its intensity and signs are different per respective subject. We choose three pairwise channels Fp1-AF3, FP1-FC2, and FP1-Oz and use scatterplots to show the relationship between the two channels. As shown in Figure 2a, the location of AF3 is adjacent to FP1, the location of Oz is distant from FP1, and the distance between FP1 and FC2 is not too far or too close. The coordinates of each point are the values of the single-channel features of two channels from the same segment. Figure 2b shows that FP1 and AF3 have a positive correlation in that as the value of AF3 increases, the value of FP1 also increases. In contrast, FP1 and Oz show a negative correlation, and there is no such relationship between FP1 and FC2. In Figure 2c, although the corresponding channel is identical to that in Figure 2b, the relationship between the two channels differs. This shows that not only are adjacent channels correlated but that all channels also have some correlations. Moreover, even with the same pairwise channels, the correlation from different subjects behaves differently. As a result, to consider subject-specific variability, the correlations between all channels must be considered.

From Figure 2, we can observe some interdependencies between channels, suggesting that the use of the channel-wise feature is necessary to consider subject-specific variability and to recognize emotions from the EEG signals of the subjects more accurately. Here, the channel-wise feature represents the interdependence between two channels, as identified by the correlation between two pairs of channels.

Therefore, we use the Pearson correlation coefficient to calculate the channel-wise feature. The Pearson correlation coefficient is a measure of the linear correlation between two variables X and Y. According to the Cauchy–Schwarz inequality, it has a value between +1 and −1, where 1 is a total positive linear correlation, 0 is no linear correlation, and −1 is a total negative linear correlation [26].

These properties of the Pearson correlation coefficient can be used to quantify how similarly the two channels change in terms of their patterns. Based on these properties, we compared every two pairwise channels among all C channels. The N channel-wise features F=F1,F2,…,FN are then computed as follows:(2)Fix,y=covSix,Siy/σSix·σSiy
where x,y∈1,C refers to the channel number and Six,Siy∈ℝK denotes the single-channel features from K consecutive segments at channel x and y. Accordingly, the channel-wise feature is described by a C×C symmetric matrix.

### 4.3. Emotional Model and Classification

To consider the longer temporal domain further, we adopt a powerful recurrent neural network (RNN) known as long short-term memory (LSTM) to model the context information of the channel-wise features.

First, we extracted N channel-wise features from the continuous time step. They can be expressed as F=F1,F2,…,FN, where Fi∈ℝC×C. Because the channel-wise features are symmetric, we only use flattened data from the upper triangle of the channel-wise features. The size of the flattened upper triangle of channel-wise features are C×C−1÷2×N and input the flattened vectors to a two-layer stacked LSTM. The hidden sequence of the first LSTM layer is input into the second LSTM layer. Accordingly, each layer has N LSTM units, and only the output from the last time step in the second layer is fed into the fully connected layer. Because the values of the channel-wise features are between −1 and 1, our model does not need a batch normalization layer, and only the dropout layer is used in the fully connected layer. The number of nodes in the last fully connected layer is determined by the number of emotion classes. The parameters in the LSTM and fully connected layer are trained to differentiate between emotion labels.

## 5. Experiments

Our goal is to develop an accurate cross-subject EEG-based emotion recognition model that considers subject-specific variability. To do this, we presented a novel emotional model with a combination of channel-wise features and a two-layer stacked LSTM.

To verify the effectiveness of the proposed method, various experiments are conducted on well-known datasets and the results are compared with those from state-of-the-art techniques. In this section, we introduce the datasets in Section 5.1 and describe the details for the experiment setting in Section 5.2. We then present hyperparameter optimization (Section 5.3), experimental results (Section 5.4), and the effectiveness of the proposed features (Section 5.5).

### 5.1. Datasets

#### 5.1.1. DEAP

The DEAP refers to the Database for Emotion Analysis using Physiological Signals. The EEG and peripheral physiological signals of 32 healthy participants (16 males and 16 females, aged between 19 and 37) were recorded while each watched 40 one-minute-long excerpts of music videos. EEG was recorded at a sampling rate of 512 Hz using 32 active AgCl electrodes (placed according to the international 10–20 system). The following peripheral nervous system signals were recorded: GSR, respiration amplitude, skin temperature, electrocardiogram, blood volume by plethysmograph, electromyograms of Zygomaticus and Trapezius muscles, and electrooculogram (EOG). The 32-channel EEG data were downsampled to 128 Hz and EOG removal was done by filtering 4.0–45.0 Hz from the data. Participants rated each video on a discrete nine-point scale for arousal, valence, like/dislike, dominance, and familiarity [27]. We only measured EEG signals and self-assessment levels of valence and arousal in our experiments. We set rating values more than 5 as high valence/arousal and less than 5 as low valence/arousal. Figure 3 plots the rating values of valence and arousal in the DEAP. The points around valence = 5 and arousal = 5 mean that subjects feel an ambiguous emotion when watching the music video. Thus, the experimental results in DEAP are not too high in previous research.

#### 5.1.2. SEED

SEED is short for the SJTU Emotion EEG Dataset. The SEED contains 15 Chinese subjects’ (7 males and 8 females, mean aged: 23.27, std: 2.37) EEG signals recorded as they watched 15 film clips. The EEG data were downsampled to 200 Hz. A bandpass frequency filter from 0 to 75 Hz was applied. For feedback, participants were told to report their emotional reactions to each film clip by completing a questionnaire immediately after watching each clip [28]. The selected videos can be understood without explanation and elicit a single desired target emotion. Thus, in our experiments, we used the labels of trials instead of the information from the questionnaires. The emotional labels contain positive, neutral, and negative attributes.

Table 1 shows the detailed information of the DEAP and SEED. As shown in this table, the two datasets have completely different properties, such as the numbers and nationalities of the subjects, and the number of trials and the channels. They also have different sampling rates. There is also an issue with noisy labels. The music videos in the DEAP are ambiguous, such that subjects may feel different emotions when watching the same video. In contrast, each film clip in the SEED is well edited to create coherent emotion elicitations and to maximize emotional meaning. Consequently, we choose the self-assessment labels in the DEAP and the categorical labels in the SEED to reduce the number of noisy labels. When a subject starts watching a video, we think that it will take some time to stimulate an emotion. Thus, we used EEG signals after 30 s in our experiments. If we can obtain good experimental results with these two different datasets, it will sufficiently explain the excellent capabilities of our model.

### 5.2. Experiment Setting

For the two-layer stacked LSTM, we set the dimension of the hidden state in the LSTM unit as 256. We adopt RMSProp to minimize the cross-entropy loss function, with a learning rate of 0.001 and a dropout probability of 0.5. Due to the limited sizes of the two EEG datasets, we apply data augmentation to increase the diversity of the training set. As we mentioned above, the input of our model is a sequence of EEG data E∈ℝC×L×K×N that contains C channels and L×K×N data per channel. From the recorded raw EEG signal G, we set ith training data Ei=GL×K×i:L×K×i+N1:C. Thus, the overlap ratio of each two adjacent training data is N−1/N. By this method, the datasets are augmented and will represent a more comprehensive set of possible data points. Then, during the training step, we randomly retrieved a mini-batch with a size of 240. We use Tensorflow 2.0.0 (Mountain View, CA, USA) and Nvidia GeForce GTX 1660 Ti (Santa Clara, CA, USA) to train our model. We used a 10-fold cross validation strategy to evaluate the effectiveness of the E-EmotiConNet using a user-independent model. We randomly split 10-fold that the same subject and the same stimuli could be both in the training set and testing set. The accuracy of the whole system is the mean classification accuracy on the test set 10 times.

### 5.3. Hyperparameter Optimization

The hyperparameters in our model are L, K, and N; specifically, we extract the channel-wise features from K consecutive EEG segments and consider the changeability of N consecutive channel-wise features in the LSTM. For DEAP, the size of the channel-wise features is 32×32; accordingly, the dimension of the upper triangle of the channel-wise feature is 496. For SEED, the size of the channel-wise features is 62×62, and the dimension of the upper triangle of the channel-wise feature is 1891. The flattened upper triangle of channel-wise features is fed into the two-layer stacked LSTM. Hence, each layer has N LSTM units.

Emotions are related to a time sequence, implying that it is important to observe emotions from multiple time steps. The performance of the proposed system was affected by several parameters; in this case, the number of data samples in each segment L, the number of segments K and the number of time steps N. To evaluate the change of the accuracy considering such parameters, we performed the first experiment. We increased the number of data samples L, using values of 2, 5, 10, 15, 20, 25, 30, 35, 40, 45, 50, 55, and 60, and extracted the channel-wise features identically to how this was done earlier, inputting them into the LSTM. We also conducted an experiment while changing the number of segments K from 4 to 12 and changing the number of channel-wise features N from 3 to 13 to measure the relationship between emotion recognition accuracy and the number of time steps in LSTM. Figure 4 presents the accuracy of emotion recognition over two-class valence in DEAP when changing the length of the segments L, the number of segments K, and the number of channel-wise features N. The experimental results are shown in Figure 4 and it shows three important discoveries:When changing the length of the segments L, the accuracy rates of emotion recognition are similar. We use a random number to initialize the parameters of LSTM. Thus, the accuracy may change slightly, and all accuracy rates are within acceptable limits. However, as L increases, more EEG data are needed. Thus, in our model, we set the length of the segments L to 2.We can also observe from the second plot that although the accuracy rates of emotion recognition do not change greatly, the results show high accuracy in two datasets when the number of segments K is 8.For the third plot in Figure 4, the accuracy of emotion recognition decreases when the number of channel-wise features is reduced. This occurs because the EEG signal consists of sequence data and the emotions change over time, implying that it is important to observe the emotions from multiple time steps. However, too much data can also be computationally expensive. Thus, we set the number of channel-wise features N to 10.

Thus, in our model, we set the number of data samples in each segment L to 2, the number of segments K to 8 and the number of channel-wise features N to 10 to consider changes of 10 consecutive channel-wise features in the LSTM and classify the emotion accurately. Consequently, our model only requires 2×8×10 data samples in each EEG channel for emotion recognition. Since the sampling rates R in the DEAP and SEED are 128 Hz and 200 Hz, we only use 1.25 and 0.8 (L×K×N/R)  second EEG data, respectively.

### 5.4. Experiments Results

An experiment was performed to prove the effectiveness of the presented channel-wise features and the two-layer stacked LSTM for cross-subject emotion classification. For DEAP, the proposed method achieved accuracy rates of 98.93% and 99.10% over the two-class classification of valence and arousal, respectively. Moreover, our model achieves high accuracy of 98.60% for four-class emotion classification (high valence high arousal, high valence low arousal, low valence high arousal, and low valence low arousal). The four-class classification model can classify valence and arousal simultaneously, meaning that there is no need to train two models separately. It also can reduce by half the number of parameters. For the SEED, the proposed method achieved an accuracy of 99.63% over three-class (positive, neutral, negative) emotion classification. Although the two datasets are different from each other, our model shows high accuracy on both datasets. This proves the robustness of the proposed model. Figure 5 shows the confusion matrices of the experiment result. Although the labels of EEG data are unbalanced, we observe that our model can recognize all the emotions correctly.

The results with the presented channel-wise features and two-layer stacked LSTM are compared with certain EEG-based emotion recognition models in Table 2 and Table 3. Wen et al. found novel convolutional neural networks for emotion recognition for the DEAP [8]. Yang et al. reported an emotion recognition system with a combination of CNN-based features and LSTM-based features [13]. Their system shows high accuracy rates of 90.80% and 91.03% on valence and arousal, respectively, but it is user-dependent in that a new model should be generated for each user. Tripathi et al. extracted nine specific values of single channels as features, with these features then being fed into a CNN [21]. Although the model achieves corresponding accuracy rates of 81.406% and 73.36% on valence and arousal classification, it may be difficult for a user to wait 63 s for the collection of the EEG signals. Wang et al. used a 3D convolutional neural network on 4-s EEG signals for emotion recognition for the DEAP [14]. Yang et al. used a combination of 10 EEG features and developed a cross-subject emotion recognition model that integrated the significance test/sequential backward selection and the support vector machine (ST-SBSSVM) [10]. Gupta et al. used the flexible analytic wavelet transform (FAWT) [22], testing their models for both the DEAP and SEED and showing accuracy rates for the DEAP below 80%, while also achieving nearly 90% accuracy for the SEED. Li. Y et al. used region and global features to develop a user-dependent emotion recognition model [29] and Li. X et al. combined 18 EEG features and test the performance on SEED. Our model achieves state-of-the-art classification rates of 98.93% and 99.10%, respectively, for two-class valence and arousal for the DEAP and shows the accuracy of 99.63% for three-class classification for the SEED. It can prove that the proposed channel-wise features and two-layer stacked LSTM can significantly improve the average recognition accuracy.

### 5.5. Effectiveness of the Proposed Features

#### 5.5.1. Effectiveness of the Channel-Wise Features

The examples of the channel-wise features are shown in Figure 6. As shown in Figure 6, the correlation is defined on two pairs of channels, and it is shown in different colors depending on the strength of the correlation. For a strong positive correlation, the corresponding cell is shown in green, while for a strong negative correlation, it is shown in red. Otherwise, for a weak correlation, the cell is white. The size of channel-wise features from SEED and DEAP are 62×62 and 32×32, respectively, since there are 62 and 32 EEG channels in the datasets.

The channel-wise features from the SEED were extracted from Subject 1, 2, 3, and 4 when they were stimulated by positive, neutral, and negative emotions. Moreover, the channel-wise features from the DEAP were extracted from Subject 1, 2, 3, and 4 when they were stimulated by high valence high arousal (HVHA), high valence low arousal (HVLA), low valence high arousal (LVHA), and low valence low arousal (LVLA) emotions. Through visualization, we discovered that although people may experience the same stimuli when watching the same video, the channel-wise features varied from person to person. Moreover, all channel-wise features of an individual from different stimuli had similar patterns. This result demonstrated that the presented channel-wise feature could adequately describe the uniqueness of the respective individuals’ EEG signals.

When comparing the channel-wise feature with the existing method, it has some advantages. Channel-wise features have many excellent properties. First, no parameters are required when extracting channel-wise features. Thus, no training steps are needed and the calculation speed is fast. Second, unlike previous CNN-based methods, which consider only adjacent EEG channels, channel-wise features calculate the interdependency of every two pairs of channels to consider the subject-specific variability factor.

Due to these useful properties, inputting the channel-wise features into the model can filter the bias of subject-specific variability and ensure good performance by the user-independent emotion recognition model.

#### 5.5.2. Effectiveness of the Emotional Features

Repurposing a pre-trained model in transfer learning tasks can reduce the training time and increase accuracy. Thus, to explore the effectiveness of learned emotional features, we used a scatter plot to visualize the output vectors from the last time step in the second LSTM layer. The emotional features consist of 256 dimensions, since the hidden state in the LSTM unit has 256 nodes. We used principal component analysis (PCA) to reduce the dimension of emotional features to two. Figure 7 shows the scatter plot of the dimension-reduced emotional features from the DEAP. The results agree with our observation in the following aspects: (1) emotional features coincide with corresponding emotions and can be classified using a simple method such as clustering or SVM; (2) our trained model can be used in transfer learning tasks such as intention detection or depression prediction.

## 6. Conclusions

In this paper, we proposed a novel cross-subject EEG-based emotion recognition model that uses a combination of channel-wise features and two-layer stacked LSTM. Our model considers subject-specific variability and reduces the noise automatically to achieve high recognition accuracy. We tested our model on two publicly available datasets, the DEAP and SEED. The accuracy rates for two-class valence and arousal in the DEAP were 98.93% and 99.10%, respectively, and the accuracy for three-class valence in the SEED was 99.63%, demonstrating that the proposed model outperforms the state-of-the-art EEG-based cross-subject emotion recognition model.

Our model can be used in the brain-computer interface (BCI) area and the channel-wise features can be used in other EEG-based tasks, such as motor imagery detection to reduce subject-specific variability. Although our model can recognize multiple subjects’ emotions, it is not easy to apply to a new subject whose data are not included in the training set. Thus, training time is needed to make another model for a new group of subjects. We will solve this problem in future works.

## Figures and Tables

**Figure 1 sensors-20-06719-f001:**
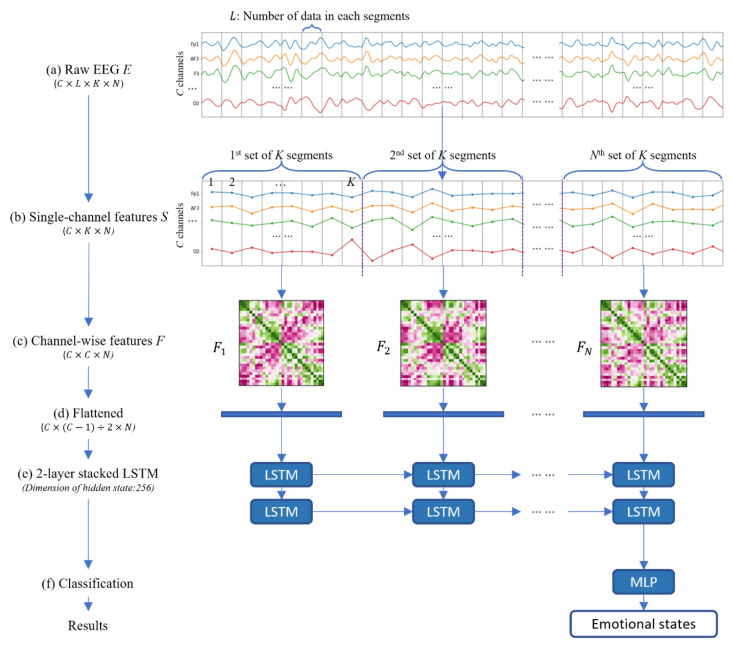
The architecture of our model. (**a**) Raw electroencephalogram (EEG) signals E. (**b**) Single-channel features S. (**c**) Channel-wise features F. (**d**) Flattened vector of channel-wise features. (**e**) Architecture of two-layer stacked Long Short-Term Memory (LSTM). (**f**) Emotion classification using Multi-Layer Perceptron (MLP).

**Figure 2 sensors-20-06719-f002:**
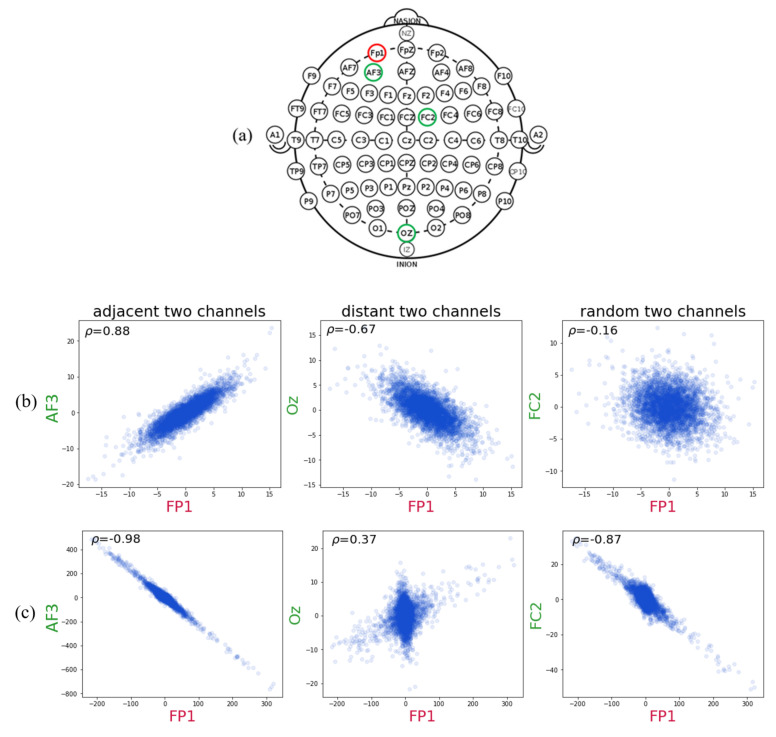
Some of the pairwise correlation between channels. (**a**) International 10–20 system for EEG electrode placement; the location of Fp1 is circled in red, AF3, FC2 and Oz are circled in green. (**b**) DEAP Subject 1, (**c**) DEAP Subject 2, (**d**) DEAP Subject 3, (**e**) DEAP Subject 4.

**Figure 3 sensors-20-06719-f003:**
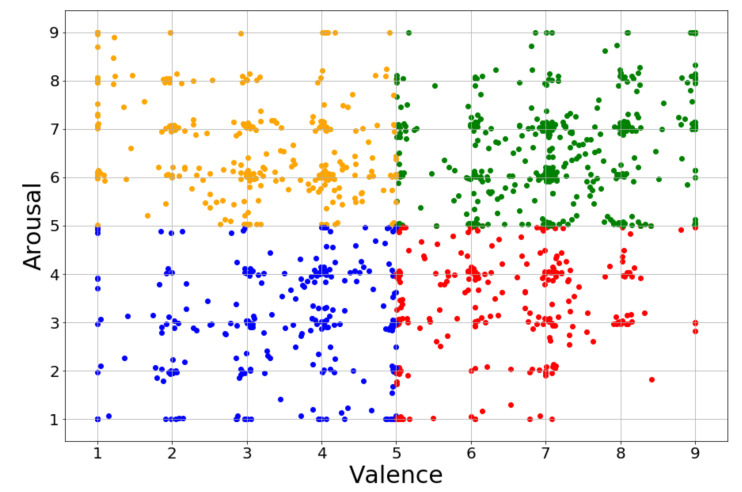
The scatter plot of rating values high valence high arousal (HVHA; green), high valence low arousal (HVLA; red), low valence high arousal (LVHA; yellow), and low valence low arousal (LVLA; blue) in the DEAP.

**Figure 4 sensors-20-06719-f004:**
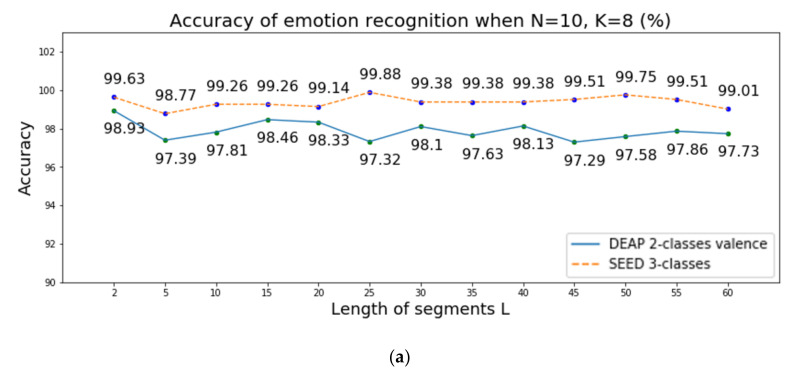
Accuracy of emotion recognition when change the parameters (**a**) L, (**b**) K, and (**c**) N.

**Figure 5 sensors-20-06719-f005:**
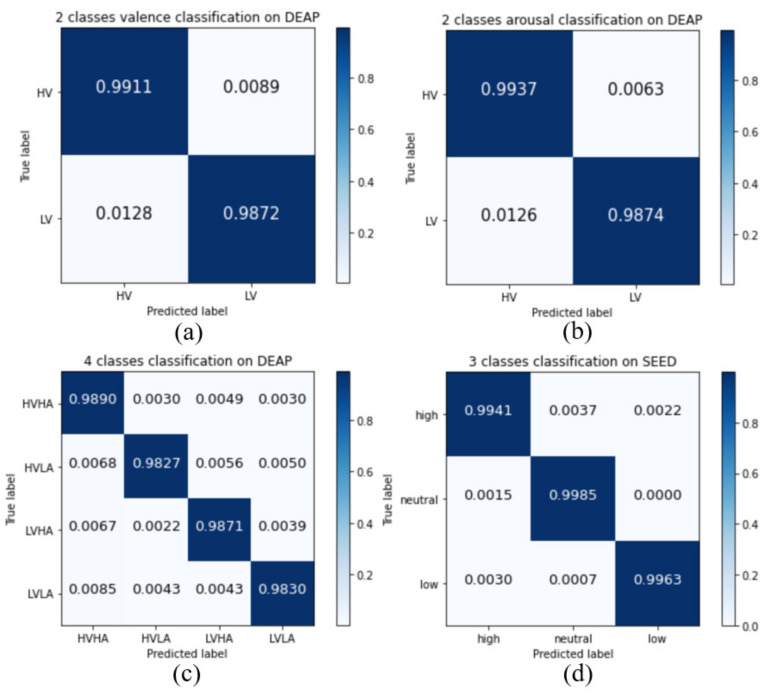
Confusion matrices of (**a**) two-class valence, (**b**) two-class arousal, (**c**) four-class classification for the DEAP, and (**d**) three-class classification for the SEED.

**Figure 6 sensors-20-06719-f006:**
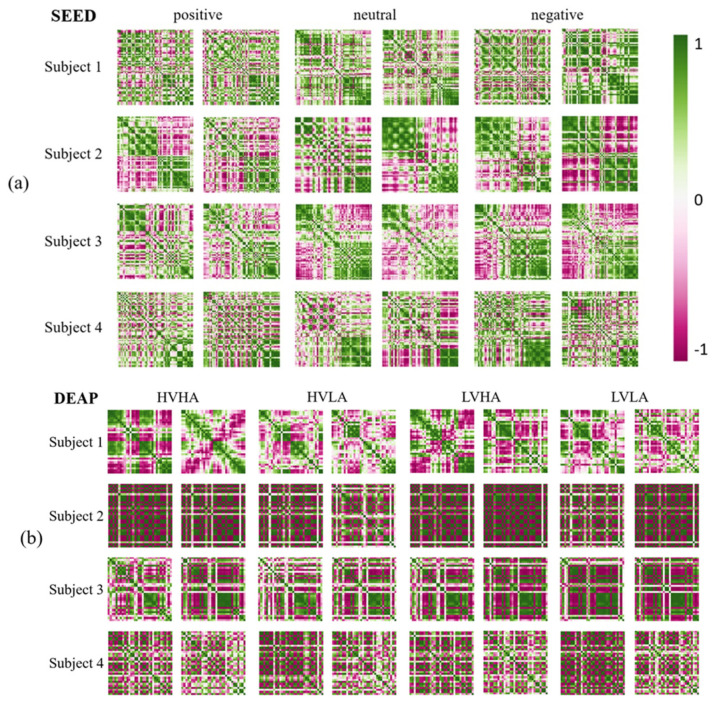
Visualization of channel-wise features and comparison between different subjects and trials. Channel-wise features are from (**a**) the SEED and (**b**) the DEAP.

**Figure 7 sensors-20-06719-f007:**
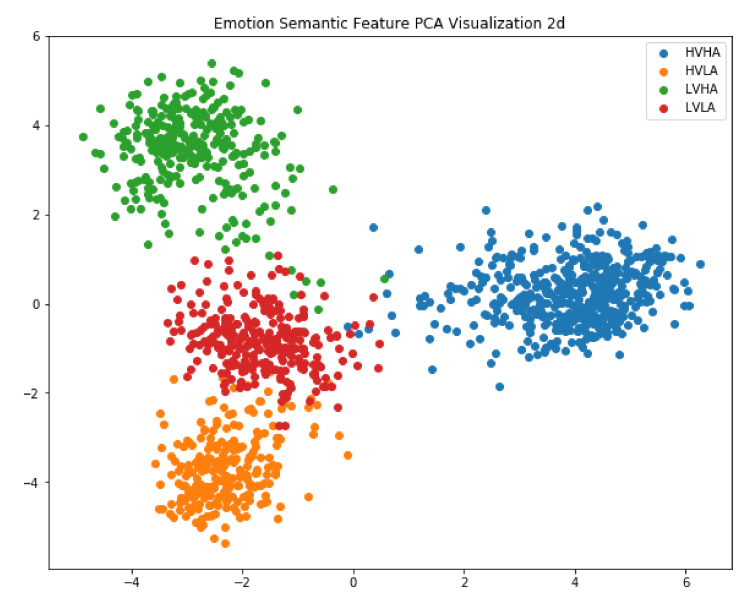
Visualization of LSTM features. The features were dimension-reduced by principal component analysis (PCA).

**Table 1 sensors-20-06719-t001:** Comparison of the two datasets.

	DEAP	SEED
Subjects	32 (Western)	15 (Eastern)
Trials	40 (63 s/trial)	15 (240 s/trial)
Channels	32	62
Sampling rate	128 Hz	200 Hz
Labels	Valence, Arousal (self-assessment)	Positive, Neutral, Negative (categorical)

**Table 2 sensors-20-06719-t002:** Comparison between classification accuracy rates of our models and previous studies for two-class (high, low) valence and arousal classification of the DEAP.

User-Dependent/Independent	Papers	Features	Length	Accuracy	
				Valence	Arousal
User-dependent	Wen et al. [8] 2017	CNN-based feature	1 s	77.98%	72.98%
	Yang et al. [13] 2018	CNN LSTM based feature	1 s	90.80%	91.03%
User-independent	Tripathi et al. [21] 2016	CNN-based feature	63 s	81.406%	73.36%
	Wang et al. [14] 2018	3D CNN-based feature	4 s	72.1%	73.1%
	Yang et al. [10] 2019	Combination of 10 EEG features	63 s	72%	-
	Gupta et al. [22] 2019	Flexible analytic wavelet transform (FAWT)	1 s	79.99%	79.95%
	Ours	Channel-wise features	1.25 s	98.93%	99.10%

**Table 3 sensors-20-06719-t003:** Comparison between classification accuracy rates of our models and previous studies for two-class (positive, neutral, negative) classification of the SEED.

User-Dependent/Independent	Papers	Features	Length	Accuracy
User-dependent	Li, Y et al. [29] 2019	Region and global features	–	88.90%
User-independent	Li, X et al. [19] 2018	Combination of 18 EEG features	4 min	83.33%
	Yang et al. [10] 2019	Combination of 10 EEG features	4 min	89%
	Gupta et al. [22] 2019	Flexible analytic wavelet transform	1 s	90.48%
	Ours	Channel-wise features	0.8 s	99.63%

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
