# Peer review of "Interpretable Cross-Subject EEG-Based Emotion Recognition Using Channel-Wise Features†"

_sensors, 2020, doi:10.3390/s20236719_

Round 1

Reviewer 1 Report

General comments:

  • This study proposes a system based on EEG for emotion recognition. This research is a relevant topic, and the results achieved are promising. The current manuscript should be organized better, providing more technical details. I recommend to present the methodology and results separately. Then, the authors should create a new section Results to present their achievements and comparison with respect to the state-of-the-art. Please see below some comments and suggestions.

Abstract

  •  Please replace the phrase "spatial interaction between channels," in line 14 pp. 1 by "spatial connectivity between brain regions". Correct it along all manuscript.
  • Replace "subjective variability" or "subjectivity of an individual's brainwaves" by "subject-specific variability".

1. Introduction

  • The authors should discussed other related works using thermography and motion capture system for emotion recognition and body expression recognition, respectively.      - Rusli, Nazreen, et al. "Implementation of Wavelet Analysis on Thermal Images for Affective States Recognition of Children With Autism Spectrum Disorder." IEEE Access 8 (2020): 120818-120834. - Andrea Kleinsmith and Nadia Bianchi-Berthouze. Affective body expression
    perception and recognition: A survey. Affective Computing, IEEE
    Transactions on, 4(1):15–33, 2013.
  • Please replace the phrase "spatial interactions of different EEG channels...." in line 70 pp. 2  by "spatial connectivity of different brain regions ...." Also make a similar action in line 87 pp. 2 "spatial interaction of all EEG channels", and other parts of this manuscript (ex lines 104-105 pp. 3).
  • Check the incomplete sentence in line 76 pp. 2 ".... temporal features simultaneously [14]. By deep learning,"
  • Modified slightly the sentence in line 111 pp.3 as ......  proposed channel-wise features can reduce "negative effect due to"  noise and subjectivity.
  • I suggest to remove or change the text in lines 62-65 pp. 2, as it is repeated in section 2. Related Works (see lines 156-159 pp. 4)

2. Related Works

  •  Please check the sentence in lines 122-125 pp.4 3, mainly  the highlight text of this phrase "...... to combine the spatial information and temporal information. of extract from raw signals using deep 124 learning techniques directly."
  • Please cite in the lines 131-134 the works: Pfurtscheller, Gert, and FH Lopes Da Silva. "Event-related EEG/MEG synchronization and desynchronization: basic principles." Clinical neurophysiology 110.11 (1999): 1842-1857. Costa, Á., Iáñez, E., Úbeda, A., Hortal, E., Del-Ama, A. J., Gil-Agudo, A., & Azorín, J. M. (2016). Decoding the attentional demands of gait through EEG gamma band features. PLoS one, 11(4), e0154136.
  • Please check the phrase mainly the highlight text: The most used bandpass filter were the Fourier transform [7] , wavelet transform [8]. I suggest to modify the sentence by "The most used method to compute power spectral feature are ......."
  • In sections 2.1 and 2.2 the authors should specify which studies used user-dependent model or/and user-independent model for better comparison, and improve the reader understanding. I think that could be important to specify the window size used to process EEG signals.

3. Overview of the Proposed Method

  • Improve the quality of Figure 1. Some texts are small, and also the first two graphics showing the raw EEG and single-channels features can be modified to present less segments and better details (channel labels, length of segments, number of segments, time T (second or minute?)). For example, Figure 2 can be integrated to Figure 1. Furthermore, add labels as (a), (b), and among others in Figure 1 to facilitate the reader understanding.
  • Reading the current section 3, it is not easy to understand the mathematical notation used for raw EEG data E ∈ ℜ C×L×K×N, specifically the variable N. Please check it in lines 207-208 pp. 5. After reading subsection 4.1 Preprocessing is possible to understand what's means N. Thus, I strongly recommend to move all related texts and Figure 1 of this section 3. Overview of the Proposed Method to section 4. Proposed method. Also provide the overview using general ideas through more comprehensive texts, avoiding details that can be better explained later to be more easily understood by readers.

4. Proposed method

  • Rewrite this sentence (see line 221 pp. 6) "Therefore, a one-minute (?=60) EEG signal of an individual has a data size of 32×128×60", mainly specifying that the raw EEG is partitioned into epochs of 1 s. Could the authors to cite previous works using epochs of 1 s? In section 2. Related Works the authors comment some previous research using window sizes between 3 and 12 s.
  • Also rewrite the second paragraph on lines 223-232, making clear what's means each variable. Currently it is a little confuse. Please specify the defined segment length L for this study.
  • Figure 2 looks redundant, as Figure 1 is showing a similar information. Please consider to remove it. Other alternative is to move Figure 2 and its related text to section Results.
  • Subsection 4.2 Channel-wise features is not only explaining the proposed method, as the authors are presenting some results, such as Figures 3 and 4 and their related text. I recommend to move these figures and their related to section Results. Please also check in Figure 3 the label corresponding to AF3. Also specify in Figure 3 the period of time analyzed and its corresponding emotion. Make a similar action for Figure 4, and explain in the title of Figure 4 what's means HVHA, HVLA, LVHA, and LVLA. This information is useful for future works, reproducibility, and readers understanding.
  • Please use only for Subsection 4.2 Channel-wise features the text on lines 273-284 pp. 8.

5. Experiments

  • In section 5.1.1 DEAP provide more details about EEG acquisition, such as EEG equipment, REF and GND electrode locations, frequency range used for acquisition, experiment setup (room temperature). As mentioned by the authors, EEG signals were downsampled at 128 Hz, but other pre-processing should be mentioned, such as EOG removal, filtering from 4.0-45.0 Hz, spatial filtering by Common Average Reference (CAR). Please check the link sharing the DEAP dataset and the reference "DEAP: A Database for Emotion Analysis using Physiological Signals", S. Koelstra, C. Muehl, M. Soleymani, J.-S. Lee, A. Yazdani, T. Ebrahimi, T. Pun, A. Nijholt, I. Patras, IEEE Transactions on Affective Computing, Special Issue on Naturalistic Affect Resources for System Building and Evaluation, in press.
  • Could the authors to provide for both datasets more information about the participants, such as age and health condition?
  • Make clear in Table 1 the period of time per trial for each dataset.
  • Please provide more details (segment length and overlapping) in this sentence (see lines 375-376 pp. 11) "We used 6 overlapped segments of EEG data from one trial and label these six segments with the same labels as the original. Is the length ? in samples or seconds? Please specify it on lines 393-394 pp. 11, "..... using values of 2, 5, 10, 15, 20, 25, 30, 35, 40, 45, 50, 55, 393 and 60,....". Make a similar action on line 404 pp. 11 "Thus, in our model, we set the length of the segments ? to 2." and in Figure 6 specifying L (also see x-axis of the top plot).
  • Move Figures 5 and 6 with their related text to section Results. I suggest to add in Figure 6 some labels, such as (a), (b), and (c) to identify each plot, explaining the performance for each setup.
  • Please rewrite this sentence (lines 416-417 pp. 12) "The sampling rates in DEAP and SEED are 128Hz and 200Hz; therefore, we only use 1.25 and 0.8 second EEG data, respectively.". It is a little confuse.
  • Present together subsections 5.4 and 5.5 in section Results. Tables 3 and 4 from subsection 5.5 can be moved to section Results, while the other texts commenting the state-of-the-art may be moved to section Discussion.
  • Improve the quality of Figure 7 and show the results using confusion matrices with values in percent. Also use a unique scale for the color bar, and therefore for all confusion matrices. Table 2 can be discarded after after following my suggestions for Figure 7.
  • This study shows the performance obtained using the pairwise correlation between channels. I encourage the authors to also evaluate their approach using only the average EEG signal corresponding to electrodes located on each one of the following regions: prefrontal, frontal lobe, temporal lobe, motor cortex, parietal lobe, and occipital lobe.  
  • Could the authors to verify if there is some relationship between channels or brain regions correlations and the video rates given by the participants?

6. Discussion

  • Subsections 6.1 and 6.2 looks like results. Subsection 6.1presents a test to prove that the proposed channel-wise features are robust to noise. Although the authors used Gaussian random noise for evaluation, this test is not enough to support their claim, as EEG is affected by physiological and non-physiological artifacts. Then, I do not know if the DEAP dataset is appropriate for this test, as it was preprocessed for removing undesirable artifacts and noise. Also, EEG data collected for emotion recognition during other conditions, such as walking may add several kinds of artifacts. I suggest to present in future works the analysis of subsection 6.1. Similarly, subsection 6.2 can be removed. Notice that Figure 9 is missed. Please avoid along all manuscript this kinds of phrases: "the proposed model showed robust results even with noise".
  • This section should be rewrite. Some texts in the current subsection 5.5 Comparison, which are commenting the state-of-the-art may be moved to section Discussion. In addition, other related studies can be discussed.
  • Provide advantages and limitations of this approach.

7. Conclusion

  • Comment practical applications and future works.

Author Response

We uploaded the answer sheet. Please see the attachment. The first part is the answers to your comments and the second part is the modified manuscript.

Reviewer 2 Report

Although the topic of the manuscript is interesting, I am sure that its experiments are methodologically incorrect and the conclusions are positively biased. Although it is not a proof, it is highly suspicious that the manuscript obtains more than 99% accuracy, which strongly overcomes other approaches reported in the literature. E.g. there is a big gap compared to many methods summarized in Table 7 of  Alarcao, Soraia M., and Manuel J. Fonseca. "Emotions recognition using EEG signals: A survey." IEEE Transactions on Affective Computing 10.3 (2017): 374-393.

The authors should redesign their experiments and especially the accuracy estimation method (testing) and repeat the experiments. Moreover, it is not clearly explained why the method should be more subject-independent than the other methods. 

Here I explain the details of my concerns:

The article states that its objective is to fight an interpersonal variability and create a user-independent classification of EEG signals to classes corresponding to emotional states. Authors point out the role of the representation of the inputs.

The correlations between all channels are used as inputs to LSTM network, which handles the temporal context and automatically learns subject-independent features.

Comment 1: Basic idea of the method is not clear

It is stated that the correlation coefficients are robust to subjectivity. I must say that although this statement is repeated multiple-times, I do not understand how is this hypothesis grounded. Lines 295-303 seems to describe the phenomenon that even goes against this hypothesis. I agree with this axiom:

” a good feature should have a similar pattern for one class and different properties for different classes”

However, the authors observe that “although people may experience the same stimuli when watching the same video, the channel-wise features vary from person to person”, which means that the patterns for one class are not similar, they highly differ between subjects. This coincides with the first part of the axiom and says that the channel-wise features are not good features for subject-independent classification. Further, authors observe that “all channel-wise features of an individual from different stimuli have similar patterns”. This coincides with the second part of the axiom. The channel-wise features thus seem to be an exact opposite of a good feature. On the other hand, authors say that “channel wise features are capable of complementarily handling subjectivity and noise”. It is not described and clear, how they are able to handle the subjectivity. My opinion is that the LSTM network could transform the sequences of those feature vectors to a good feature (subject-independent), but it is not described how. The main idea of the paper is thus hidden for me and not understandable. The following description is not enough: “Due to these useful properties, inputting the channel-wise features into the model can reduce the bias of subjectivity and ensure good performance by the user-independent emotion recognition model.” Moreover, I also do not get the following sentence, which should be explained more clearly: “Channel-wise features calculate the interdependency of every two pairs of channels to consider the subjectivity factor”

Comment 2: Performance estimation method seems to be positively biased

In section 5.2, only a short statement about 10-fold-cross-validation is provided. In the most common settings, the distribution of the data instances into the folds is being performed randomly. This cannot be done in case of the temporal context aware models like LSTM. It is therefore not clear, how the data are split into the folds. For the example of DEAP data, there are 32 subjects (participants) and 40 videos for each.

1 - For an unbiased evaluation of the classification method, it must not happen that data instances from the same subject and the same stimulus are both in the training and testing set. From my experience, this can cause an extremely positive bias of the performance estimate. A testing instance will have a close-in-time training instance, which will be naturally very similar from the EEG signal point of view, which hides a real (un)correlation between EEG signal and classes. For example, if there will be some type of trend, unfiltered during the signal preprocessing, this will be especially strong.

2- For an unbiased evaluation of the user-independency of the classification method, it must even not happen that data instances from the same subject are both in the training and testing set. Otherwise, there is again a positive bias.

For those reasons, I would expect and suggest a subject based type of CV. For example, there can be 32-folds and each fold can contain only data from one subject. This is most probably not the case of the presented experiments. There are 10 folds, which can mean e.g. that there are data from 3 subjects in the first 9 folds and the last fold contains data from 5 subject (3*9+5=32). I have however a strong doubt about this. I recommend the authors to use a correct method and clearly describe details of the cross-validation setting since an incorrect usage of this can lead to absolutely incorrect conclusions (it is my personal experience).

Comment 3: Accuracy is not clearly defined

Authors use the accuracy as the performance criterion. The accuracy should be however clearly defined in the article text, especially for the four-class-case.

Comment 4: The tuning causes additional positive bias of the final accuracy estimate

In the tuning described in section 5.3, it seems that the same cross-validation error was used as the one reporting in the following “Experimental results” section. Such a tuning cause another serious positive bias of the final accuracy estimate. The tuning should be performed on a set, which is independent from the testing set. The data handling is not described in the manuscript, however I can prove that it is incorrect by comparing the accuracies for the  tuned combination L=2, K=8, N=10 à98.93 (DEAP), 99.63% (SEED) and the final accuracies presented in Table 2. The fact that they are the same proves my hypothesis that the performance evaluation is methodologically incorrect.

Some other issues:

Lines 207-213 should be rewritten and made more clear. Although finally I understood, the paragraph is not easy to read.

Line 252: “The data in Figure 3 are from subject 1, 2, 3 and 4 in DEAP dataset”, the is an inconsistency with Caption of Figure 3, where the subjects are: “(b) subject 1, (c) subject 2, (d) subject 4, (e) subject 14”.

Lines 304-313: The description is not clear:

“No parameters are required when extracting channel-wise features. Thus, no training steps are needed and the calculation speed is fast.” – which training is mentioned? Feature extraction mostly does not contain training steps.

Author Response

(The authors gave the same response as above.)

Reviewer 3 Report

The authors cited references without mentioning the advantages and drawbacks. It would be better if the authors indicate some criticism regarding each cited method. Then, motivations of the present work and the gap that is filled by this research paper should be highlighted. Therefore, I suggest rewriting the related work section.

The authors could also make some improvements in the English used in the article. Avoid over usage of similar words, such as "propose" . The authors should enrich their language with alternative expressions.

 It is not clear whether the whole data was used for the training or not.

Dose the algorithm at any point require human supervision, what is need for making the algorithm fully automatic.

Please outline some of the practical applications and limitations of the work

Do the authors have any plans to make the code and data-set open-source?

Author Response

(The authors gave the same response as above.)

Round 2

Reviewer 1 Report

I would like to thank the authors for attending all my comments and suggestions. I have still a major concern about the methodology used for evaluation, which is presented in my comment v. Also, the revised manuscript should be better organized. Please see below my comments and suggestions:

  1. Please define CWF on lines 213 pp. 5.
  2. Check the sentence "In this work, the LSTM is employed as a filter to remove subject specific patterns." (see lines 215 pp.5).
  3. I recommend to replace the term "functional connectivity" by "spatial connectivity", as this study analyzes spatial connectivity between brain regions.
  4. Give more details in this phrase (see lines 372-373 pp.11) "We used 6 overlapped segments of EEG data 372 from one trial and label these six segments with the same labels as the original". Make clear the segment length and percent of overlapping between segments. I noted that the segment length is informed in Tables 3 and 4. Also, section 5.5 Ablation studies explains later the best values for L, K, and N. I recommend to organize better section 5. Experiment to improve the reader understanding.
  5. DEAP dataset contains a total of 40 trials with 63 s of duration each one. SEED dataset contains a total of 15 trials with 240 s of duration each one. However, only one period of 1.25 s (for DEAP) and 0.8 s (for SEED) is analyzed from each trial. The authors should explain why segments over the full trial are not used for training and testing. Please note that this methodology is not usual. For example, Gupta et. al. [10] 2019 extracted one second epochs from the last 30 seconds of the recorded EEG signals, and similar strategies over a full trial had been used by Wang et. al. [14] 2018, Gupta et. al. [10], Yang et. al. [16] 2018, and Wen et. al. [6] 2017. Please review the methodology used, considering the aforementioned studies for an appropriate comparison with the state-of-the-art.

Author Response

We uploaded the answer sheet. Please see the attachment. 

Reviewer 3 Report

Authors have addressed my comments. 

Author Response

Thank you for your comments.

Round 3

Reviewer 1 Report

I would like to thank the authors for attending all my comments and suggestions. The revised manuscript have improved, but there is still some aspect to clarify. Please see below my comments and suggestions:

  1. As mentioned in my first revision, the authors cannot support that this approach is robust to noise. Please update the related affirmations throughout all text. See line 18 pp.1,  lines 90-91 pp.2 : (line 18) "subject-specific variability and noise. The channel-wise features are then fed to two-layer stacked";  (lines 90-91) "We include visualizations of channel-wise features and show that the channel-wise features are robust to noise and subject-specific variability."
  2. Use capital letter when referring to each section on the lines 11-118 pp. 3. For example, "... in Section 4", "... in Section 5", "... in Section 6", and "... in Section 7". Make a similar action when referring to specific subjects (ex. Subject 1 or Subjects 1, 2, 3 and 4) in the manuscript (see line 267 pp. 7 and line 287 pp. 8).
  3. Replace "developing a classification methods" by ''developing classification methods" on the line 121 pp. 3. Please proofreading carefully the manuscript to correct other minor grammar mistakes.
  4. Review the mathematical notations. Use italic letter for variables, and bold letter when referring to vectors, such as E (see line 249 pp. 6), X and Y (see line 294 pp. 6), F (see line 299 and 310 pp. 9).
  5. The authors define N of two forms: 1) see lines 227-228 "? is the number of channel-wise features"; 2) see line 395"... the number of time steps ?." I think that the term "channel-wise features" to define N is a little confuse, as I am understanding that N represents each set of consecutive K segments.
  6. I am still a little confuse reading the lines 372-376 pp. 11, mainly for these parts: "We extracted ?×?×(?+5)/? second epochs ..." and "The size of window is ?×?×?/? and the stride is ?×?/?." Please check the term (N+5). The authors could update Figure 1 making clear that N represents a set of K segments, as analyzing Figure 1, it is not easy to understand why N is used to represent the window length. ? refers the number of channel-wise features.  Could the authors to update Figure 1 if it is possible? Rewrite this sentence (see lines 375-376) "For example, we first extract 15 (?=10) consequent channel-wise features [?1,?2,…,?15]. Seeing Figure 1 the authors first extracted EEG epochs. Could the authors to provide more details in this paragraph (lines 372-379) using some texts (R=128 for DEAP and R= 200 for SEED) from the lines 241-246 pp. 6? It could benefit the reader understanding.
  7. Please note that Figure 1 shows the length (L) of each segment in second. Then, it is confuse to read these texts: (see line 226)" ? is number of EEG data samples"; (line 394) "the number of data samples in each segment ?"; (lines 396) "We increased the number of data samples ?, using values of 2, 5, 10, 15, 20, 25, 30, 35, 40, 45, 50, 55, and 60". Please update Figures 1 (a) and 1(b), and review carefully the sentence on the lines 396-397 "We increased the number of data samples ?, using values of 2, 5, 10, 15, 20, 25, 30, 35, 40, 45, 50, 55, and 60 .......". Are these values in seconds? Please make it clear on the text and Figure 4a.

Author Response

We revised the manuscript and uploaded the answer sheet. Please see the attachment.
